# A Comparative Study on the Presence and Reversibility of Subclinical Arterial Damage in HCV-Infected Individuals and Matched Controls

**DOI:** 10.3390/v15061374

**Published:** 2023-06-15

**Authors:** Theodoros Androutsakos, Dimitrios Mouziouras, Stamatia Katelani, Mina Psichogiou, Petros P. Sfikakis, Athanase D. Protogerou, Antonios A. Argyris

**Affiliations:** 1Clinic/Laboratory of Pathophysiology, School of Medicine, National and Kapodistrian University of Athens, 11527 Athens, Greece; skatelani@yahoo.gr (S.K.); aprotog@med.uoa.gr (A.D.P.); 2Department of Gastroenterology, Laiko General Hospital, School of Medicine, National and Kapodistrian University Athens, 11527 Athens, Greece; dmouziouras@med.uoa.gr; 3Cardiovascular Prevention and Research Unit, Clinic/Laboratory of Pathophysiology, School of Medicine, National and Kapodistrian University of Athens, 11527 Athens, Greece; anargyr@med.uoa.gr; 4First Department of Internal Medicine, Laiko General Hospital, School of Medicine, National and Kapodistrian University Athens, 11527 Athens, Greece; mpsichog@med.uoa.gr; 5First Department of Propaedeutic Medicine, Laiko General Hospital, School of Medicine, National and Kapodistrian University Athens, 11527 Athens, Greece; psfikakis@med.uoa.gr

**Keywords:** HCV, direct-acting antivirals, arterial stiffness, atheromatosis, carotid plaques

## Abstract

**Background:** The arterial pathology and mechanisms of increased cardiovascular disease (CVD) risk in HCV-infected individuals are not yet clear. The aim of this study was to identify types of arterial pathology in treatment-naive chronic HCV patients and to test their reversibility after successful treatment. **Methods:** Consecutive, never-treated, HCV-infected patients were compared with age and CVD-related risk factors, matched controls, healthy individuals (HI), patients with rheumatoid arthritis (RA) and people living with HIV (PLWH), in terms of arterial stiffening by pulse wave velocity, arterial atheromatosis/hypertrophy by carotid plaques/intima-media thickness and impaired pressure wave reflections by augmentation index. After three months of sustained virological response (SVR) administered using direct-acting antivirals, vascular examination was repeated in HCV-infected patients to test drug and viral-elimination effect in subclinical CVD. **Results:** Thirty HCV patients were examined at baseline; fourteen of them were re-examined post-SVR. Compared with HI, HCV patients had significantly more plaques, which is similar to that of RA patients and the PLWH group. No other differences were found in all other vascular biomarkers, and regression among HCV patients also revealed no differences 3 months post-SVR. **Conclusions:** Accelerated atheromatosis, rather than arterial stiffening, arterial remodeling and peripheral impaired hemodynamics is the underlying pathology leading to increased CVD risk in HCV patients.

## 1. Introduction

Chronic hepatitis C virus (HCV) infection constitutes a major global health issue. It is estimated that more than 70 million individuals are infected with HCV worldwide, with at least 400,000 deaths occurring each year [1,2]. However, the landscape of chronic HCV infection treatment has changed substantially in the last decade with the emergence of direct-acting antiviral drugs (DAAs) as a first-line treatment against HCV infection, leading to more than 95% sustained virological response (SVR) in all patients, irrespective of the presence of cirrhosis, comorbidities or genotype [3,4,5].

Chronic HCV infection is characterized by a variety of extrahepatic manifestations (EHMs); among them, cardiovascular disease (CVD) is the most important as patients with chronic HCV exhibit increased cardiovascular mortality [6,7]. Interestingly, the arterial pathology and mechanisms linked to increased cardiovascular events are not yet clear [7]. The use of DAAs and the subsequent eradication of HCV seem to lead to a decrease in CVD-related events [8,9], however, studies focusing on the early steps of CVD by measuring subclinical arterial damage show contradictory results [10,11,12].

The aim of the present study was to identify evidence of accelerated subclinical arterial damage in treatment-naive patients with chronic HCV, without established CVD. To this end, we investigated three different types of arterial damage: (i) arterial stiffening by carotid to femoral pulse wave velocity (cfPWV), (ii) arterial atheromatosis/hypertrophy by common carotid plaques/intima-media thickness (IMT) and (iii) impaired pressure wave reflections by augmentation index (AIx75). In order to test the hypothesis of accelerated subclinical arterial disease in HCV-infected individuals, we used the following comparators: (i) a matched group of apparently healthy individuals (HI) and (ii) two other matched groups, consisting of patients with rheumatoid arthritis (RA) and people living with HIV (PLWH); both these groups have been extensively investigated in the literature due to the presence of accelerated subclinical arterial damage. Finally, in order to investigate the possibility of arterial damage reversibility in HCV patients after successful eradication of HCV, the above-mentioned vascular examination was repeated 3 months post-SVR.

## 2. Materials and Methods

### 2.1. Study Population—Chronic HCV Patients

Study population consisted of consecutive patients with treatment-naive, chronic HCV infection, referred for evaluation and treatment to the outpatient hepatology clinic of the Pathophysiology department of the “Laiko” General Hospital from October 2018 to November 2019. Inclusion criteria were the presence of HCV infection (defined as detectable HCV RNA in peripheral blood) and age > 18 years; exclusion criteria were the inability to receive treatment (due to poor compliance, concurrent alcoholism or intravenous drug use), concurrent viral hepatitis other than HCV infection and the presence of autoimmune liver disease or cancer. The study was approved by the Ethical/Scientific Committee of the “Laiko” Hospital and all participants provided informed consent according to the declaration of Helsinki.

Medical history was obtained through patients’ interviews and their medical charts. Duration of HCV infection was defined as the duration between patients’ HCV diagnosis (via PCR assessment of HCV viral load) and enrollment day. All patients underwent transient elastography before DAA initiation; liver stiffness was graded between 1 (normal liver) and 4 (cirrhotic liver) according to EASL guidelines for non-invasive assessment of HCV infection [13]. On the inclusion date, blood was drawn and thorough laboratory testing (including serum transaminases, serum total, high and low-density cholesterol and serum triglycerides levels, glycosylated hemoglobin and serum glucose levels) were performed. Type 2 diabetes mellitus (T2DM) was defined as concurrent antidiabetic treatment or levels of blood sugar ≥ 126 mg/dL or ≥ 200 mg/dL for fasting and non-fasting patients, respectively, or glycosylated hemoglobin of (HbA1c) ≥ 6.5%. Dyslipidemia was defined as concurrent lipid modifying treatment or LDL ≥ 130 mg/dL in double check at least 3 months apart according to guidelines [14,15]. Arterial hypertension was defined as systolic/diastolic blood pressure (SBP)/(DBP) ≥ 130 and/or ≥80 mmHg using 24 h ambulatory blood pressure monitoring (ABPM) or through the concurrent use of anti-hypertensive treatment [16].

Regimens used for HCV treatment followed the national and international guidelines at the time each patient was treated, after completing baseline vascular examinations [17]. SVR was defined as the absence of HCV RNA in a patient’s blood 3 months after DAA treatment, according to guidelines [17].

### 2.2. Study Matched-Control Groups

Three separate groups were used as comparators: (i) a group with HI, (ii) a group of patients with RA and (iii) a group with PLWH. The selection of RA and PLWH as non-healthy matched control groups was based on the fact that these populations have been extensively investigated in the past regarding the presence of accelerated subclinical arterial damage by our group [18,19,20,21,22,23,24], as well as by others [25,26,27,28]. All 3 control groups were CVD-free and were recruited from the outpatient CVD prevention clinic of our hospital.

For each separate control group, HCV patients were matched 1:1 for possible confounders, including age, smoking habits, history of hypertension, dyslipidemia, T2DM, body mass index (BMI), SBP during the examination.

### 2.3. Assessment of Subclinical Arterial Damage

All participants underwent vascular examination, which was performed by the same experienced physician. All participants were on steady medication for at least one month and were asked to abstain from any vasoactive medication in the morning of the vascular assessment. Blood pressure measurements were conducted in the morning (8:30–12:30 a.m.) in the supine position, after at least 10 min of rest, in a controlled room temperature (22–25 °C). A validated, automated device (Microlife Watch BP Office, Microlife AG, Widnau, Switzerland) was used; 3 branchial blood pressure measurements (with 1 min interval between each measurement) were recorded. The average of all 3 measurements, as well as the calculated mean arterial pressure (MAP) was used in the statistical analysis. cfPWV and AIx75 were assessed according to the methodology described elsewhere, using the SphygmoCor apparatus (AtCor Medical, Sydney, Australia) [29,30]. Briefly, cf-PWV was calculated by the ratio of the estimated pulse transit time and the distance travelled by the pressure wave between the two recording sites. Pressure waves were first recorded at the carotid artery and then at the femoral artery. The time delay between the two waves (transit time) was determined using registration with a simultaneously recorded ECG. Two sequential measurements of cfPWV were performed and their average value was used in the analysis. AIx was also assessed as an index of pressure wave reflections and corrected to a heart rate of 75 beats/min. Using radial applanation tonometry and a generalized transfer function applied to the non-invasively acquired peripheral signal, an aortic waveform was calculated. AIx was then generated as the augmentation pressure (systolic pressure minus the inflection pressure) divided by the pulse pressure (systolic minus diastolic pressure), expressed as a percentage [31]. Left and right IMT at the common carotid artery and atheromatic plaques at the carotid and femoral beds were assessed using B mode vascular ultrasonography (Logiq V5 Expert, GE Healthcare, Fairfield, CT, USA). IMT was measured adjacent to any plaque (if present) at the far wall of the common carotid artery in the end-diastole of the cardiac cycle, via a semi-automatic software. Atheromatic plaque was defined as local increase in the IMT by ≥50% compared to the adjacent vessel wall or a bulging to the lumen IMT ≥ 1.5 mm [30]. For all vascular biomarkers, 2 consecutive measurements were obtained and the mean value was used in the analysis.

In order to test the hypothesis of arterial damage reversibility after HCV eradication, a scheduled follow-up vascular examination was performed 3 months post-SVR in patients with HCV infection.

A validated brachial-cuff ABPM device was used for the evaluation of blood pressure in HCV patients (Mobil-O-Graph; IEM, Stolberg, Germany); out of office blood pressure measurements were also used in all other groups.

### 2.4. Statistics

All statistical analyses were performed using the STATA 13 software package [32]. Statistical significance was defined as a *p*-value of <0.05. For description purposes, frequencies distributions and medians (interquartile range) or mean plus standard deviations were used for age, MAP, heart rate, BMI and duration of HCV infection. Wilcoxon sign-rank test was performed between all comparison groups. All comparisons on the presence of atheromatic plaques were performed through McNemar chi-squared test. In multivariable linear regression, further adjustment for MAP was carried out, heart rate during the vascular examination was measured to compare left IMT, right IMT and PWV between HCV patients and their respective matched controls, and AIx5 was adjusted only for MAP. For the presence of atheromatic plaques univariable conditional logistic regression was performed, since all patients were matched. In order to further investigate the differences in vascular biomarkers among HCV patients, multivariable linear regression, as well as univariable logistic was performed. The covariates used were the ones listed above, in addition to duration of HCV infection, stage of liver fibrosis and age.

## 3. Results

### 3.1. HCV Baseline Measurement

Overall, 29 treatment-naive patients (23 men and 6 women, mean age: 46.6 ± 10.7 years) with chronic HCV infection were included in the study; one was excluded due to concurrent arterial hypertension. Twenty-eight patients were treated with glecaprevir/pibrentasvir (GP), while one was treated with grazoprevir/elbasvir. The baseline characteristics of HCV-infected individuals are presented in Table 1. Table 2 summarizes the measured vascular biomarkers of the comparison groups.

### 3.2. HCV Patients’ Comparisons with the Matched Control Group

Effective 1:1 matching was achieved for HI, RA patients and PLWH in 29, 19 and 20 untreated patients with HCV, respectively (Table 1).

Concerning arterial stiffness, cfPWV was consistently lower in HCV patients, as compared to PLWH in unadjusted models, but not HI; however, after adjustment for possible confounders, no differences were observed between HCV and any of the three control groups (Table 2 and Table 3).

Regarding atheromatosis, plaque occurrence was almost double in HCV patients compared to the HI control group, reaching statistical significance (Table 3); on the contrary, no differences were observed between HCV patients and RA or PLWH. HCV patients had significantly lower left IMT compared to matched RA patients in unadjusted models, with this difference being attenuated in adjusted models. In addition, no differences between HCV and any of the other control groups regarding left or right IMT were observed (Table 2 and Table 3).

Finally, Aix75 was found to be statistically lower in HCV patients compared to RA patients, however, after adjustment for MAP during the vascular examination, this difference was eliminated (Table 2 and Table 3). No other differences between HCV patients and HI, as well as PLHW were observed (Table 2 and Table 3).

### 3.3. HCV Patients Follow-Up Visit 3 Months Post-SVR

All treated patients achieved SVR; however only 14 of them (all having received GP treatment) presented for the vascular follow-up re-examination 3 months post-SVR (Table 4). No differences were observed in any of the vascular indices among these 14 patients at the follow-up visit. In unadjusted models, PWV was significantly lower at 3 months; nevertheless, after adjustment for MAP differences between visits, no treatment effect was observed (Table 5).

### 3.4. HCV Patients’ In-Group Analysis

Data on duration of HCV infection and stage of liver fibrosis were available for 22 out of the 30 HCV patients enlisted (Table 1). The duration of HCV infection without treatment was found to have a positive association with all the vascular biomarkers under investigation, apart from PWV; however, statistical significance was not reached for any of them (Table 6). As far as liver fibrosis is concerned, it seemed that advanced fibrosis was associated with the elevation in all vascular biomarkers, except PWV, even though statistical significance was not reached in any of them; the only exception was the higher values of left IMT in patients with liver cirrhosis (grade 4 fibrosis) (Table 6).

## 4. Discussion

The CVD complications of chronic HCV infection have gained much attention in recent years, since a variety of studies and meta-analyses suggest that a significant—and probably causative—correlation between the two aforementioned diseases exists, with HCV-infected individuals showing a higher probability for coronary artery disease (CAD), stroke, acute myocardial injury and CVD-related death [33,34,35,36]. In one of the largest meta-analysis, Petta S. et al. showed that HCV-infected individuals showed an odds ratio of 1.65 for overall CVD deaths and a 2-fold higher risk for the presence of carotid plaques [33]. However, the exact pathogenesis of these phenomena still remains elusive.

This single-center prospective study in patients with treatment-naive, CVD-free, chronic HCV infection, aimed at (i) identifying the presence of early steps in different types of arterial damage (atheromatosis, arterial hypertrophy, arterial stiffening and impaired pressure wave reflections) using control groups at low or high CVD risk, as well as aimed at (ii) assessing the hypothesis of arterial damage reversibility in HCV patients after SVR. Clear evidence of accelerated subclinical atheromatosis, with higher prevalence of atheromatic plaques in HCV-infected individuals—but not of other types of arterial damage—was observed.

The present results suggest that accelerated subclinical atheromatosis might represent the very early step in the sequel of CVD development in treatment-naive HCV-infected patients, and this finding that is in line with existing evidence. On the contrary, all other types of arterial damage, as evaluated by the gold standard biomarkers applied herein, seem to be preserved, at least in the early stages of CVD development. It is plausible to suggest that HCV virus directly affects the endothelial layer by promoting atherogenesis, without affecting the elastic compartment of the medial layer, thus sparing arterial hypertrophy (no increase in IMT in plaque—free arterial segments) and arterial stiffening (no increase in PWV). The underlying mechanism for this predilection might be related to the hepatic and systemic inflammation induced by HCV, resulting in increased levels of proatherogenic cytokines and chemokines, mainly interleukin-6 (IL-6), tumor necrosis factor alpha (TNF-a) and fibrinogen and an imbalance between pro- and anti-inflammatory cytokines, leading to chronic inflammation and oxidative stress and subsequent endothelial damage of blood vessels [37,38].

On the other hand, our analysis among HCV patients provides evidence that HCV infection has most probably some effect in arterial hypertrophy as well, since IMT shows a statistically significant elevation for every year of chronic HCV infection in both carotids, while excessive liver fibrosis caused by the virus also seems to affect arterial thickening.

Our findings are partly in concordance with the first pivotal Japanese study which revealed the possible link between hepatitis C virus and subclinical vascular damage [39,40]. In these studies, HCV-infected patients were found to have increased atheromatic burden, translated in statistically significant differences in both the prevalence of atheromatic plaques and IMT, compared to controls. It is important to mention that measurement methods used can influence the IMT values, and this can describe the discrepancies observed between studies. More specifically, many researchers prefer to use the mean value of multiple mean or maximum IMT measurements, while there is also deviation in the carotid segment where IMT is evaluated (carotid bulb vs. common carotid artery vs. internal carotid artery). Furthermore, some prefer to include atheromatic plaques in the IMT measurements leading to greater values reported [41].

In the present study, plaque burden in HCV-infected individuals was twice as much when compared with HI and equal to that observed in RA patients and PLWH. It has been previously described [18,19,20,21,22,23,24,25,26,27,28] that both these conditions (RA and PLWH) have almost double—than their control groups—the burden of subclinical atheromatosis. RA patients have higher incidence of CVD and almost 40% of all deaths are attributed to CVD complications [42]. Moreover, CVD risk in RA is underestimated by the currently available international CVD risk score [43] and testing for subclinical carotid plaque presence has been proposed in order to optimize risk reclassification in RA patients [44]. Likewise, PLWH seem to have increased CVD risk which is underestimated by the available CVD risk scores [45], however, international recommendations regarding CVD risk reclassification have not yet incorporated arterial indices/risk modifiers.

No evidence of early arterial stiffening or arterial remodeling (hypertrophy and impaired peripheral pressure wave reflections) in patients with HCV infection was found in this study. Any changes in arterial properties during the follow-up period in HCV-infected patients was most probably related to MAP changes rather than arterial wall modifications. We have previously described that the abovementioned arterial properties, although reported in the literature to be impaired in populations with chronic inflammation (e.g., RA), are most probably only transiently impaired and reversible upon effective treatment and regression of inflammation [45,46,47]. Of note, plaque regression was not examined in the present study since 6 months is per se not an adequate time to study such an effect.

Our findings suggest that HCV virus per se rarely causes vascular damage, but most probably enhances the effect of well-known risk factors, such as age. Moreover, a larger duration of infection and hepatic cirrhosis seems to be accompanied with increased atherogenesis, but decreased arterial stiffness, expressed through PWV, though in our cohort, statistical significance was not reached. The decreased arterial stiffness could be explained by the splanchnic and peripheral arterial vasodilation found in patients with cirrhosis, leading to significant alterations in vascular properties [48].

The first novelty and strength of the present study lies on the simultaneous and comprehensive evaluation of multiple types of subclinical arterial damage (i.e. arterial stiffness, arterial remodeling, atheromatosis and impaired pressure wave reflections) at multiple arterial sites, with the use state-of-the art vascular biomarkers, that are well accepted as CVD risk modifiers [49,50,51,52,53,54,55,56]. The second novelty of the study lies in the use of multiple control groups, including—for the first time—two well-studied, high CVD risk populations (i.e., RA and PLWH). The main limitations of the present study are the relatively small sample size, the absence of long follow-up, as well as the big percentage of dropout during the follow-up period at three months post-SVR, making it mandatory to validate our findings in large, well designed, long-term studies in the future.

## 5. Conclusions

The present findings suggest that atheromatosis, rather than arterial stiffening, arterial remodeling and peripheral impaired hemodynamics, is the underlying pathophysiological mechanism leading to increased CVD risk in HCV patients. Future large cohort studies should be undertaken in order to confirm our findings and optimize CVD prevention strategies.

## Figures and Tables

**Table 1 viruses-15-01374-t001:** Baseline characteristics of the treatment-naive HCV patients and of the matched HI, RA and PLWH groups.

	HCV n = 29	Controls n = 29	RA n = 19	PLWH n = 20
Age (years)	46.3 ± 10.7	46.6 ± 10.5	50.1 ± 910	46.1 ± 11.3
Male sex [n (%)]	23 (79.3)	23 (79.3)	14 (774)	18 (90)
Hypertension [n (%)]	0 (0)	0 (0)	0 (0)	0 (0)
Diabetes mellitus [n (%)]	0 (0)	0 (0)	0 (0)	0 (0)
Dyslipidemia [n (%)]	3 (10.3)	3 (10.3)	1 (5.2)	1 (5)
Cardiovascular disease [n (%)]	1 (3.4)	1 (3.4)	2 (10.4)	0 (0)
Smokers [n (%)]	18 (62.1)	18 (62.1)	10 52.6)	14 (70)
Ex-smokers [n (%)]	5 (17.2)	5 (17.2)	4 (21)	3 (15)
Duration of HCV infection (years)	12.4 ± 9.3			
Liver fibrosis grading				
Grade 1	13 (43)			
Grade 2	5 (17)			
Grade 3	2 (7)			
Grade 4	2 (7)			
BMI (kg/m^2^)	25.± 4.2	24.5 ± 2.8	26.1 ± 4.4	25.5 ± 4.5
Heart rate (beats/min)	65 ± 7	65 ± 11	65 ± 8	66 ± 8
Mean arterial pressure (mmHg)	87 ± 8.1	89.5 ± 8.2	87.5 ± 8.8	87 ± 8.3
Dyslipidemia medication [n (%)]	0 (0)	3 (10)	1 (5)	0 (0)

*p*: non-significant for all comparisons of matched HI, RA and PLWH groups with HCV group. Abbreviations: BMI: Body mass index; HCV: Hepatitis C virus; HI: Healthy individuals; RA: Rheumatoid Arthritis; PLWH: People living with HIV.

**Table 2 viruses-15-01374-t002:** Unadjusted comparison of vascular biomarkers between treatment-naive HCV patients and control groups.

Vascular Biomarkers	HCV (n = 29)	HI (n = 29)	HCV (n = 19)	RA (n = 19)	HCV (n = 20)	PLWH (n = 20)
PWV (m/s)	6.9 ± 1.0	7.4 ± 1.5	6.9 ± 0.93	7.8± 1.9	6.8 ± 0.9	7.2 ± 0.8 *
Right IMT (mm)	0.59 ± 0.13	0.59 ± 0.11	0.63 ± 0.14	0.75 ± 0.14	0.62 ± 0.15	0.65 ± 0.13
Left IMT (mm)	0.63 ± 0.15	0.66 ± 0.11	0.65 ± 0.15	0.75 ± 0.14 *	0.59 ± 0.14	0.59 ± 0.06
AIx75 (%)	20.3 ± 11.0	21.1 ± 12	23 ± 9.2	28.9 ± 10.2	18.1 ± 11.4	17.5 ± 13.9
Plaques [n (%)]	17 (58.6)	10 (34.5)	8 (42)	11 (58)	11 (55)	10 (50)

Wilcoxon sign-rank test or Mc Neymar chi-squared test were performed as appropriate. * *p* < 0.05 for comparisons with HCV group. Abbreviations: HI: Healthy individuals; AIx75: Augmentation index adjusted for 75 bpm; HCV: Hepatitis C virus; IMT: Intima media thickness; PLWH: People Living With HIV; PWV: Pulse wave velocity; RA: Rheumatoid arthritis.

**Table 3 viruses-15-01374-t003:** Adjusted comparison of vascular biomarkers between treatment-naive HCV patients and matched groups.

Vascular Biomarkers	b	Odds Ratio	*p*
PWV (m/s)			
HCV vs. HI	−0.4	-	0.21
HCV vs. RA	−0.97	-	0.07
HCV vs. PLWH	−0.39	-	0.13
Left IMT (mm)			
HCV vs. HI	0.01	-	0.87
HCV vs. RA	−0.07	-	0.13
HCV vs. PLWH	−0.01	-	0.73
Right IMT (mm)			
HCV vs. HI	0.02	-	0.55
HCV vs. RA	−0.07	-	0.17
HCV vs. PLWH	−0.01	-	0.98
AIx75 (%)			
HCV vs. HI	1.3	-	0.66
HCV vs. RA	−5.2	-	0.14
HCV vs. PLWH	0.1	-	0.98
Atheromatic plaques			
HCV vs. HI	1.1	2.9	0.04
HCV vs. RA	0.21	1.24	0.74
HCV vs. PLWH	0.69	2	0.42

Linear regression models adjusted for blood pressure and heart rate, except for Aix75, which is adjusted only for blood pressure. Logistic regression model was unadjusted for any covariate. Abbreviations: HI: healthy individuals; AIx75: augmentation index adjusted for 75 bpm; HCV: hepatitis C virus; IMT: intimal-medial thickness; PWV: pulse wave velocity; RA: rheumatoid arthritis; PLWH: people living with HIV.

**Table 4 viruses-15-01374-t004:** Comparison of characteristics among HCV patients at baseline and 3 months post-SVR.

	HCV Baseline n = 14	HCV 3 Months after SVR n = 14
Age (years)	45.5 ± 10.4	45.5 ± 10.4
Male sex [n (%)]	12 (86)	12 (86)
Hypertension [n (%)]	0 (0)	0 (0)
Diabetes mellitus [n (%)]	0 (0)	0 (0)
Dyslipidemia [n (%)]	1 (7)	1 (7)
Cardiovascular disease [n (%)]	1 (7)	1 (7)
Smokers [n (%)]	7 (50)	7 (50)
Ex-smokers [n (%)]	3 (21)	3 (21)
BMI (kg/m^2^)	25.2 ± 3.5	28.8 ± 3.5
Heart rate (beats/min)	66 ± 7	64 ± 8
Mean arterial pressure (mmHg)	82 ± 10.1	83.6 ± 13.7
Antihypertensive medication [n (%)]	0 (0)	0 (0)
Dyslipidemia medication [n (%)]	0 (0)	0 (0)
Diabetes mellitus medication [n (%)]	0 (0)	0 (0)

Abbreviations: BMI: Body mass index; HCV: Hepatitis C virus, SVR: Sustained virological response.

**Table 5 viruses-15-01374-t005:** Comparison of vascular biomarkers in HCV patients at baseline (before treatment initiation) and follow-up (3 months after sustained virological response).

Vascular Biomarkers	Baseline	Follow-Up	*p* *	*p* **
PWV (m/s)	6.74 ± 0.83	6.38 ± 1.04	0.02	0.08
Left IMT (mm)	0.62 ± 0.14	0.59 ± 0.11	0.78	0.34
Right IMT (mm)	0.57 ± 0.09	0.58 ± 0.1	0.57	0.89
AIx75 (%)	19.4 ± 10.7	22.4 ± 12.3	0.61	0.57

*: Wilcoxon sign-rank test. **: multivariate linear regression model. Abbreviations: AIx75: augmentation index adjusted for 75 bpm; HCV: hepatitis C virus; IMT: intimal-medial thickness; PWV: pulse wave velocity.

**Table 6 viruses-15-01374-t006:** Comparison of vascular biomarkers among HCV patients: regression models.

Covariates	PWV (m/s)	Right IMT (mm)	Left IMT (mm)	AIx75 (%)	Plaques
HCV Duration (years)	−0.03	0.004	0.005	0.02	0.97
Liver fibrosis grading					
Grade 1 (reference)					
Grade 2	0.003	0.006	0.03	2.9	0.2
Grade 3	−1.4	−0.01	0.0001	2.3	0.28
Grade 4	−0.79	0.01	0.21	9.4	1.001

*p*: non-significant for all comparisons. Abbreviations: AIx75: augmentation index adjusted for 75 bpm; HCV: hepatitis C virus; IMT: intimal-medial thickness; PWV: pulse wave velocity.

## Data Availability

All data regarding this article are available upon reasonable request.

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
