# Peer review of "A Comparative Study on the Presence and Reversibility of Subclinical Arterial Damage in HCV-Infected Individuals and Matched Controls"

_viruses, 2023, doi:10.3390/v15061374_

Round 1
Reviewer 1 Report
The abstract and title was a bit confusing to read. I understand that you are using patients with rheumatoid arthritis and HIV as comparators for two other chronic illnesses to say that the changes observed are specific to HCV (if any). That is a good design but all of that does not have to go in to the title. Keep it short and mention HCV patients only. Regarding abstract it needs to be reworded also for clarity specially the methods part.
What is the "training" received by the persons doing these measurements - some of these measurements are subjective to interpretation (vascxular ultrasonography) and it will be reassuring if the same person took all measurements or two people independently agreed on the observations
This article needs proof-reading by a native English speaker - For example, in line 114, it should be medication, not medicament.
For Table 1, p values should be given for comparisons
Were those with hypertension and diabetes excluded or matched with others with a similar diagnosis, if its the latter you need a proper sample size calculation as there are so many confounders and I feel 20-30 individuals is inadequate to make the conclusions. I see that there is only one patient with HT in many groups (and no diabetes). It would have been more convient to exclude those with HT in terms of sample size calculation.
If Section 3.4 is going to be discussed in results, this needs appropriate mention in introduction and methods
In table 1 there are two columns stating "HCV matched to RA" and RA matched to HCV" (same for HIV). This is confusing. You are only matching HCV to RA and not vice versa. Also the results in subsequent sections discuss HCV matching to RA and HIV only (which is correct). I suggest simplifying Table 1 and deleting uneccessary columns
Table 6 is difficult to interpret without p values. If none were significant you can add a footnote to say p>0.05
This article needs proof-reading by a native English speaker - For example, in line 114, it should be medication, not medicament.
There are several other mistakes like that throughout which I do not have time to highlight one by one
Author Response
- The abstract and title was a bit confusing to read. I understand that you are using patients with rheumatoid arthritis and HIV as comparators for two other chronic illnesses to say that the changes observed are specific to HCV (if any). That is a good design but all of that does not have to go in to the title. Keep it short and mention HCV patients only. Regarding abstract it needs to be reworded also for clarity specially the methods part.
Authors’ reply: We thank the reviewer for the constructive comments. We agree that the title is too long and may be confusing, so we have shortened it according to suggestion. We have also updated the abstract, especially the methods section. We hope our abstract and title are now more clear.
- What is the "training" received by the persons doing these measurements - some of these measurements are subjective to interpretation (vascxular ultrasonography) and it will be reassuring if the same person took all measurements or two people independently agreed on the observations
Authors reply: Thank you very much for your comment. We have updated the methods section 2.3, clarifying your question,as follows: “All participants underwent vascular examination by the same, experienced physician”.
- This article needs proof-reading by a native English speaker - For example, in line 114, it should be medication, not medicament.
Authors reply: Thank you very much for your comment. The article has undergone proof-reading and hope that it has been greatly improved.
- For Table 1, p values should be given for comparisons
Authors reply: Thank you very much. Since all groups are matched with the HCV patient group, all comparisons are non-significant. This has been added as a footnote of table 1.
- Were those with hypertension and diabetes excluded or matched with others with a similar diagnosis, if its the latter you need a proper sample size calculation as there are so many confounders and I feel 20-30 individuals is inadequate to make the conclusions. I see that there is only one patient with HT in many groups (and no diabetes). It would have been more convient to exclude those with HT in terms of sample size calculation.
Authors reply: We thank the reviewer for the comment. In order to decrease the possibility of any random effect and bias due to confounding factors - even though we have implemented a thorough matching process - we have excluded the patient with arterial hypertension from all comparisons. In line with that we have updated relevant tables and results.
- If Section 3.4 is going to be discussed in results, this needs appropriate mention in introduction and methods
Authors reply:Thank you very much for your remark. We had already given details of how HCV disease duration was calculated, as well as how transient elastography was used in “study matched-control groups” in “Materials and methods section”; however to make it more clear we have moved it to “Study population: chronic HCV patients” section and have also added appropriate reference.
- In table 1 there are two columns stating "HCV matched to RA" and RA matched to HCV" (same for HIV). This is confusing. You are only matching HCV to RA and not vice versa. Also the results in subsequent sections discuss HCV matching to RA and HIV only (which is correct). I suggest simplifying Table 1 and deleting uneccessary columns
Authors reply: Thank you very much for your comment. We understand this is confusing so we have deleted the unnecessary columns.
- Table 6 is difficult to interpret without p values. If none were significant you can add a footnote to say p>0.05
Authors reply: Thank you very much for your remark. P values have now been added in this table, too.
- Comments on the Quality of English Language
This article needs proof-reading by a native English speaker - For example, in line 114, it should be medication, not medicament.
There are several other mistakes like that throughout which I do not have time to highlight one by one
Authors reply: Thank you very much for your comment. The article has undergone proof-reading and hope that it has been greatly improved.
Reviewer 2 Report
The paper shows interesting data concerning the possibility of HCV being responsible for arterial diseases. Unfortunately, the number of cases is low, and the follow-up is too short. The absence of a correlation between the length of the HCV infection and arterial damage argues against this hypothesis. Since many factors can influence the evolution of cardiovascular diseases (like physical exercise, lifestyles, smoking, genetic factors and so on), authors should be more cautious in their conclusions.
The English should be improved.
Accept after minor changes
Author Response
- The paper shows interesting data concerning the possibility of HCV being responsible for arterial diseases. Unfortunately, the number of cases is low, and the follow-up is too short. The absence of a correlation between the length of the HCV infection and arterial damage argues against this hypothesis. Since many factors can influence the evolution of cardiovascular diseases (like physical exercise, lifestyles, smoking, genetic factors and so on), authors should be more cautious in their conclusions.
Authors reply: Thank you very much for your comment. We totally agree that due to the low number of patients in our cohort our findings should be interpreted with caution. Apart from already adding the low number of our cohort as a limitation, we have also re-phrased the “conclusions” section, highlighting the fact that larger studies are needed in order to verify our findings.
- The English should be improved.
Authors reply: Thank you very much for your comment. The article has undergone proof-reading and hope that it has been greatly improved
Reviewer 3 Report
This manuscript reports the results of a single-center prospective study where patients with HCV infection were compared with healthy individuals (matched patients without cardiovascular diseases). A second comparison regarded HCV-infected patients versus patients with rheumatoid arthritis; finally, the authors conducted another comparison (HCV infected individuals versus HIV-infected patients).
Table 3 suggests that the number of plaques is greater in HCV-infected patients in comparison with controls; the IMT (right and left intima media thickness) is lower in HCV-infected patients than in those suffering from rheumatoid arthritis.
The results obtained in the study appear difficult to understand to the busy reader; in addition, the multiple comparisons between subgroups of patients that have been reported regard small sized cohorts.
Author Response
This manuscript reports the results of a single-center prospective study where patients with HCV infection were compared with healthy individuals (matched patients without cardiovascular diseases). A second comparison regarded HCV-infected patients versus patients with rheumatoid arthritis; finally, the authors conducted another comparison (HCV infected individuals versus HIV-infected patients).
Table 3 suggests that the number of plaques is greater in HCV-infected patients in comparison with controls; the IMT (right and left intima media thickness) is lower in HCV-infected patients than in those suffering from rheumatoid arthritis.
The results obtained in the study appear difficult to understand to the busy reader; in addition, the multiple comparisons between subgroups of patients that have been reported regard small sized cohorts
Authors reply: Thank you very much for your comment. We agree that our manuscript could be troublesome to read, since it contains much technical information; however we feel that it brings something novel to the HCV-related CVD-risk and hope it will help those who choose to read it to flourish their knowledge regarding this issue.
As far as the small size of our groups, we totally agree with you and have already stated that as our major limitation.
Round 2
Reviewer 3 Report
This is a cohort study with 29 patients with chronic HCV infection who were included in the study. The authors evaluated vascular biomarkers between patients with chronic HCV and controls such as healthy individuals (HI), patients with rheumatoid arthritis (RA), and HIV positive individuals. Various types of arterial damage were evaluated. The investigators did not find significant differences between the study and control groups with regard to arterial stiffening (carotid to femoral pulse wave velocity), right/left intima media thickness (IMT), and pressure wave reflections. There was significant difference with regard to arterial atheromatosis as the number of plaques was much greater among HCV-infected individuals compared with controls (HI). These results have been appropriately discussed in the Section Discussion of the manuscript.
It remains unclear if the patients with HCV infection completed antiviral therapy with DAAs before the initial phases of the study. In other words, I do not understand if the baseline parameters shown in Table 1 (demographic, biochemical, clinical parameters) and Table 2 (vascular biomarkers) have been obtained after SVR or after completing antiviral therapy or before. I do believe that this piece of information is important to the busy reader.
I suggest to change the title of Table 4 as it is not appropriate.
Author Response
We thank the reviewer for the useful comments. Patients with HCV infection had not received any treatment before the initiation of the study. So, baseline data in tables 1 and 2 were obtained before any treatment was offered to these patients. A second, follow-up visit was scheduled 3 months after achievement of SVR. This is stated in abstract (“The aim of this study was to identify types of arterial pathology in treatment naïve, chronic HCV-patients and to test their reversibility after successful treatment”) and methods (“Study population consisted of consecutive patients with treatment naïve, chronic HCV infection…”). However, in order to make this more clear, we have updated the information in all sections of the manuscript (abstract, introduction, methods, results and tables). We have also modified the title of table 4.